# From ToyBox Study to eToyBox: Advancing Childhood Obesity Reduction in Malaysian Kindergartens

**DOI:** 10.3390/ijerph20166614

**Published:** 2023-08-20

**Authors:** Sue Reeves, Bee Koon Poh, Yi Ting Chong, Julia Ai Cheng Lee, Whye Lian Cheah, Yatiman Noor Hafizah, Georgia Nelson, Abd Talib Ruzita, Denise Koh, Carolyn Summerbell, Cecilia A. Essau, Edward Leigh Gibson

**Affiliations:** 1School of Life and Health Sciences, University of Roehampton, London SW15 4JD, UK; s.reeves@roehampton.ac.uk; 2Centre for Community Health Studies (ReaCH), Faculty of Health Sciences, Universiti Kebangsaan Malaysia (UKM), Kuala Lumpur 50300, Malaysia; chongyiting.16@gmail.com (Y.T.C.); hafizahyatiman@ukm.edu.my (Y.N.H.); rzt@ukm.edu.my (A.T.R.); 3Faculty of Cognitive Sciences and Human Development, Universiti Malaysia Sarawak (UNIMAS), Kota Samarahan 94300, Sarawak, Malaysia; aclee@unimas.my (J.A.C.L.); georgialivannelson@gmail.com (G.N.); 4Faculty of Medicine and Health Sciences, Universiti Malaysia Sarawak (UNIMAS), Kota Samarahan 94300, Sarawak, Malaysia; wlcheah@unimas.my; 5Centre for Education and Community Wellbeing, Faculty of Education, Universiti Kebangsaan Malaysia (UKM), Bangi 43600, Selangor Dahrul Ehsan, Malaysia; denise.koh@ukm.edu.my; 6Department of Sport and Exercise Sciences, Durham University, Durham DH1 3LE, UK; carolyn.summerbell@durham.ac.uk; 7School of Psychology, University of Roehampton, London SW15 4JD, UK; c.essau@roehampton.ac.uk (C.A.E.); l.gibson@roehampton.ac.uk (E.L.G.)

**Keywords:** childhood, obesity, intervention, nutrition, energy balance, preschool, online

## Abstract

Prevention and treatment of childhood obesity is a global concern, and in Malaysia, it is considered a national public health priority. Determinants of childhood obesity are multifactorial and include factors that directly and indirectly influence energy balance-related behaviours, including energy intake and energy expenditure. Interventions to address childhood obesity that have multiple components at different levels have been shown to be the most influential. The ToyBox-study is a childhood obesity intervention aimed at preschool-aged children and their families that had been shown to be effective in several European countries and so was chosen for adaption for the Malaysian setting. Materials were translated and adjusted for the Malaysian context and audience and implemented in kindergartens in Peninsular Malaysia and Sarawak. However, during the COVID-19 pandemic and lockdown, teaching transitioned to being online. This brought an opportunity to reach a wider audience and consider the long-term sustainability of the intervention, and thus eToybox was born. eToybox aims to bring support for healthy energy balance behaviours directly to the teachers, into kindergartens and homes, to encourage families to be active and eat healthily, and prevent or reduce obesity. Through online innovation, the Toybox Study Malaysia programme has been expanded to enhance its potential to impact the promotion of healthy lifestyles among preschoolers and their families, highlighting the importance of a holistic approach to preventing and treating childhood obesity in Malaysia.

## 1. Introduction

The World Health Organisation [1] has stated that the prevention and treatment of childhood obesity is a global concern because of the direct impact on child health and wellbeing. Since the consequences of childhood obesity can be lifelong and result in greater risk of developing non-communicable diseases (NCDs) in adulthood [2,3], this emphasises the importance of taking an early life-course approach [2]. 

Globally, between 1975 and 2016, the prevalence of being overweight or living with obesity increased from 4% to 18% in children aged 5–19 years [4]. Furthermore, in children under 5 years of age, it was estimated that 41 million children were living with obesity or were overweight [5]. In Malaysia, a survey of a nationally representative sample of children aged 6 months to 12 years (*n* = 3542) reported that 11.8% were living with obesity and 9.8% were overweight [6]. Furthermore, when comparing preschool children aged 4–6 years in urban and rural areas, the prevalence of overweight and obesity were found to be 16% and 17%, respectively [6]. More recently, The National Health and Morbidity Survey [7] reported the prevalence of being overweight and obese among children under five years old as 5.6%. As such, the prevention of childhood obesity is considered a national public health priority [8,9].

The determinants of childhood obesity are complex and multifactorial and comprise of biological, genetic, environmental, and behavioural factors [10]; these factors may also directly and indirectly influence energy balance-related behaviours, which include energy intake (diet) and energy expenditure (physical activity) [11]. A Cochrane review concluded that interventions with multiple components were the most effective in treating obesity among preschool children [12]. Reviews of studies aimed at preventing childhood obesity have noted that these interventions should include components, such as nutrition and exercise, conducted at different levels, such as at school and at home [13,14]. The need to incorporate multiple-components strategies, such as nutrition, physical activity, and behaviour change, to treat children and adolescents with obesity was also highlighted in the American Academy of Paediatrics Clinical Practice Guideline on the treatment of children and adolescents with obesity [15]. This guideline also highlights the importance of the school environment as a setting for improving energy balance behaviours since children spend so much time at school [15].

At the time this study was initiated, there were no other known obesity prevention interventions targeting preschool children in Malaysia and that the ToyBox-study kindergarten intervention looked to be the most promising to be implemented in Malaysia as it had been shown to be effective in several European countries [16]. The aim of the ToyBox-study was to prevent early childhood obesity using an evidence-based approach to promote healthy food, drinks, and activities for preschool-aged children and their families [16,17]. The design of the ToyBox-study intervention followed the PRECEDE-PROCEED logical model [18], used intervention mapping as a framework [19], as well as incorporated theories of behaviour change [17]. The ToyBox Study Malaysia programme was an adaptation and translation of the original ToyBox-study and targeted the same four energy balance-related behaviours as the original study, namely eating healthy food, drinking water, increasing physical activity, and reducing sedentary behaviour [20]. The study was implemented in two areas, Peninsular Malaysia and Sarawak. Following the COVID-19 pandemic, ToyBox Study Malaysia was further adapted to become a web-based intervention programme. Hence, the primary objective of this paper is to present the process of adapting and implementing the ToyBox Study programme within the Malaysian context and its intervention impact and outcomes, while the secondary objective is to provide an overview of transforming the physical intervention programme into its online counterpart, the digital learning platform named eToyBox.

## 2. Methods

### 2.1. Approval for ToyBox Study Malaysia 

ToyBox Study Malaysia aimed to encourage a healthy lifestyle in Malaysian children, with the involvement of teachers and parents. The study targeted children from lower income families who attended kindergartens under the Community Development Department (KEMAS) and Ministry of Rural and Regional Development; both KEMAS and the Ministry of Rural and Regional Development gave their approval for the study. Consent was obtained from all teachers, parents, and guardians prior to the study commencing. The first phase of the ToyBox Study Malaysia ethics clearance was approved by the Research Ethics Committee of the Universiti Kebangsaan Malaysia (UKM) numbered UKM PPI/111/8/JEP-2017-658, Universiti Malaysia Sarawak (UNIMAS) numbered UNIMAS/NC-21.02/03-02 Jld.2(68), and also registered with the International Standard Randomised Controlled Trial Number (ISRCTN) Registry, the UK clinical trial database (ID ISRCTN16593038). 

### 2.2. Development of Modules 

The ToyBox Study Malaysia materials, such as the guidance modules and questionnaires, were adapted from the ToyBox-study originally conducted in Europe [17,21]. Cultural and linguistic adaptations were made to ensure the materials were appropriate for the Malaysian context and audience.

The adaptation and translation process of the modules were organised through five stages: Adaptation of the ToyBox-study modules and componentsReview of the ToyBox-study questionnaires and evaluation formForward and backward translation and proofreading of the ToyBox Study Malaysia materialsFace validationHarmonising and finalising the ToyBox Study Malaysia materials

Both modules and questionnaires underwent the adaptation process. Cultural adaptations were made through discussions in a series of workshops and meetings among the research team members and stakeholders, including the teachers and parents of the preschoolers. Focus group discussions (FGDs) were conducted using a semi-structured question guide to achieve the main objectives of the adaptation process, which were to modify the material according to the Malaysian context and culture. The FGDs were successfully conducted with preschool teachers, teaching assistants, and parent groups in Kuala Lumpur, Selangor, and Sarawak. Two meetings with KEMAS officers and preschool supervisors were conducted to provide feedback on the contents of the modules and help identify contents that were not suitable for our young children. Linguistic adaptation was performed in a standardised way by using forward and backward translation. The translation was completed independently by two Malaysians fluent in both the English and Malay languages and with backgrounds in nutrition and education, experts in intervention studies, and a professional translator. The final version of the translated materials was tested for face validity among preschool teachers and parents. Lastly, the translated materials were proofread by two independent professional proofreaders. Any differences in wording or terms used were reconciled into a single version, which the research team approved by consensus. 

Some content, including the logo, were modified to fit the Malaysian culture and context. Figure 1 shows the ToyBox Study Malaysia logo adapted from the ToyBox-study. All final materials were then validated using the face validation method by KEMAS teachers and officers and also parents of KEMAS preschoolers. Feedback from the validation process was then harmonised into those materials, and the research team finalised the modules.

### 2.3. Theory of Change Intervention Mapping 

Full day workshops to develop the Theory of Change (TOC) were conducted at both sites. Stakeholders from KEMAS (Early Childhood Education Department), comprising 13 officers, 18 preschool teachers, and 13 parents, as well as the ToyBox Study Malaysia research team, participated in the TOC workshops. One of the UK researchers, an expert in the TOC process, facilitated the discussions based on theories and conceptual frameworks related to health intervention and implementation to develop the TOC map. Ideas from participants were collated and mapped to structure the implementation of the intervention. Figure 2 displays the constructed TOC map. The TOC map was used to guide the implementation of the ToyBox Study Malaysia programme in the selected kindergartens.

### 2.4. Implementation 

ToyBox Study Malaysia was implemented over a period of 6 months in both Klang Valley, Peninsular Malaysia (sub-urban and urban areas), and Sarawak (rural areas). The intervention was implemented based on the time periods suggested in the original ToyBox-study that was conducted in Europe, i.e., for 24 weeks, with a total of 6 weeks for each of the four modules, namely healthy eating, drinking, physical activity, and reducing sedentary behaviours. A specific implementation timeline schedule was developed for implementation in Malaysia based on the academic calendar of KEMAS kindergartens. Figure 3 shows the timeline for the ToyBox Study Malaysia implementation. As the intervention kindergartens in Sarawak were in remote areas, the implementation timeline was slightly adjusted to accommodate logistical issues. 

Participants (teachers) were trained prior to the implementation through Training of Teachers (TOT) sessions conducted by the research team. A total of three TOT sessions were conducted at 2-monthly intervals throughout the entire implementation period. A teacher’s package was provided, which included a kangaroo mascot, hand and finger puppets, balls, bean bags, SukuSukuSeparuh (quarter-quarter-half) plates, water tumblers, water dispenser, and other useful materials. Printed modules were provided to the teachers for their reference. Newsletters and bulletins for parents were also given to the preschool teachers to be distributed according to the module implementation schedule throughout the intervention period. A total of 48 preschools (971 children) participated, including 22 preschools (483 children) in the intervention arm and 26 preschools (488 children) in the control arm. The control group did not receive any form of intervention during the study.

The process of adapting the ToyBox-study from the European context to Malaysia encountered few challenges, particularly concerning the indoor and outdoor activities recommended in the modules, such as physical games in class and excursions. The majority of preschools faced space constraints, making it impractical to implement many of the original games. As a result, we reviewed and selected alternative games that required less space to be carried out effectively. Additionally, activities like excursions to countryside areas, maize field labyrinths, and swimming were not viable options in the Malaysian setting, especially considering their remote locations. Therefore, we suggested that teachers take the children for a walk around the village compound instead.

Furthermore, we made modifications by adopting the Malaysia Healthy Plate, known as the “SukuSukuSeparuh” plate. This adaptation proved more effective in teaching children about healthy food choices and appropriate portion sizes when compared to the Magic Snack Plate recommended in the original modules, which mainly focused on consuming healthy foods like fruits, vegetables, bread, and cheese as snacks.

By addressing these challenges and making necessary adjustments, the ToyBox Study was successfully tailored to suit the unique needs and conditions of the Malaysian context, providing valuable insights into childhood obesity reduction strategies in Malaysian kindergartens.

## 3. Intervention Impact and Outcome Measures 

The four targeted energy balance-related behaviours were measured via the Primary Caregiver Questionnaire adapted from the European ToyBox-study [21,22], and changes pre- and post-intervention in these behaviours are shown in Table 1. The trend of change shows Toybox Malaysia Study increased plain water consumption (from 3.86 glasses/day pre-intervention to 3.9 glasses/day post-intervention) and reduced the intake of sweet snacks “kueh” and sandwich cookies in the intervention group (from 0.82 portion/day to 0.75 portion/day post-intervention) as compared to the control group. Participation in outdoor active play showed an increase from 56.8 min/day to 61.3 min/day post-intervention in the intervention group. Outdoor active play was also significantly higher in the intervention group compared to control group post-intervention (61.3 vs. 43.9 min/day, *p* < 0.05). Furthermore, although the amount of time children spent on sedentary screen time in both groups showed an increase, there was no significant change in the intervention group, whilst the control group observed a significant increase (from 2.46 h/day to 2.85 h/day, *p* < 0.05). These improvements in the four targeted energy balance-related behaviours may also have impacted BMI status, since there was an overall reduction in BMI z-score among overweight children and children with obesity in the intervention group at the post-intervention follow-up [23].

Teachers’ feedback on the implementation of ToyBox modules was gathered after the completion of the programme. Most teachers expressed willingness to continue implementing ToyBox modules in their teaching and felt that the modules were flexible and the activities were easy to carry out in their classrooms. Although some teachers were concerned that the modules might take too long to complete, and that it could be burdensome to their workload, most of them would still recommend the modules to their colleagues as the Toybox modules are in line with and offer interesting activities that complement the KEMAS curriculum. 

The implementation of ToyBox Study Malaysia has also increased the awareness among stakeholders, including school administrators, principals, teachers, parents, and children, on the importance of healthy lifestyle habits. Furthermore, as previously published [24], it was concluded that teachers and parents’ perspectives provided favourable support that the ToyBox preschool-based behavioural intervention demonstrated impact in the knowledge and practices regarding energy balance-related behaviours. By adopting healthier lifestyle habits, children and families could experience improved economic outcomes in the long run, with potential benefits such as reduced sick days, increased parental productivity, and enhanced income due to decreased absenteeism.

### 3.1. Process Evaluation 

A process evaluation was conducted to confirm the intervention was implemented effectively. Using the protocol of the European ToyBox-study, a process evaluation was conducted based on five indicators: recruitment, retention, dosage, fidelity, and satisfaction. The findings of the process evaluation reported that the feedback from the teachers and parents was encouraging, supporting the notion that the ToyBox Study Malaysia intervention programme has been successful. A detailed report on this has recently been published [25].

### 3.2. Sustainability of Obesity Prevention Intervention

There is increasing concern that sustainability should be a crucial part of programme implementation. However, advancements on the issue of sustainability have been made primarily in conceptual papers rather than empirical studies [26]. Generally, the sustainability of an intervention programme refers to the new and beneficial approaches becoming a norm in the targeted environment over time [26,27]. The capacity to sustain the changes resulting from innovative obesity prevention programmes in policy and practice is an important endeavour [24] for ensuring that long-term outcomes for the public are realised and community support is maintained [26]. Obesity prevention entails the implementation of innovative approaches to be maintained in several spheres, including the continued practise of the approaches, the changed behaviours of the participants, and the ongoing benefits to people and systems [28]. In the past, the impact of sustainability in school-based obesity intervention programmes had been examined primarily in high-income nations [29]. There were two studies carried out under the project that explored the sustainability of ToyBox Study Malaysia, with one published [24] and another still ongoing. The findings concluded that the reciprocal interactions between teachers and parents help to improve the behaviour of the children who, in turn, serve as agents of change [24]. 

In the ongoing qualitative study investigating the sustainability practices of teachers who had undergone the ToyBox Study Malaysia implementation project in 2017–2018, the teachers reported that the ToyBox Study Malaysia programme was easily adapted to fit the existing school operation and curriculum. Sustaining ToyBox Study Malaysia came with several challenges, such as the COVID-19 pandemic national lockdown in Malaysia, and thus physical teaching transitioned to online learning to avoid disease transmission. Since the reopening of the schools, time was found to be another issue because preschools operate for just half a day, so the implementation of the ToyBox programme is, therefore, restricted. Nevertheless, the teachers felt strongly that the programme should reach a wider audience at the school level as well as the community level. This led to the idea to digitise the content so programme sustainability could be achieved through eToyBox. With eToybox being online, accessibility could be made possible anywhere, anytime, and for anyone. In this way, not only do the teachers play a role in promoting a healthy lifestyle, but the parents can also help reinforce such a lifestyle at home [30]. 

## 4. Digitalising ToyBox Study Malaysia

Despite the majority of physical obesity prevention programmes positively impacting behavioural changes and health outcomes, the sustainability of their implementation faces various challenges in reaching their intended stakeholders due to time constraints, geographical limitations, and inadequate budgets [31]. Furthermore, financial and human resources are the most identified factors associated with the sustainability of interventions or expected outcomes [32]. The digitalisation of obesity prevention programmes, on the other hand, provides more accessibility, cost-effectiveness, and time flexibility [33].

An extensive body of research suggests that skill development related to content knowledge or intervention improves programme sustainability [29]. As a result, educating through digitisation is the most cost-effective way to sustain the impact of an intervention [32]. A well-designed online learning programme can offer convenience, relevance, and tailored and precise content that the targeted users need whilst still providing the same or even better knowledge and skills as in-person training [34,35]. In addition, to the accessibility and flexibility of online learning, a broader targeted audience can be reached, and more information can be disseminated with less time required. A previous study has demonstrated the benefits of online learning as a strategy for preventing obesity by enhancing knowledge, attitude, and self-efficacy [36] and improving positive health outcomes [37]. Hence, online learning could be a cost-effective intervention approach to obesity prevention since less input and contact are needed to achieve significant and sustainable effects compared to in-person intervention programmes [38]. 

Recent statistics on internet use in Malaysian adults found that internet usage by individuals increased from 89.6% in 2020 to 96.8% in 2021 (+7.2%). Meanwhile, 98.7% and 83.5% of the population utilised mobile phones and computer accounts, respectively [39]. To assess the needs of developing eToyBox, preschool teachers from different states of Malaysia were invited to evaluate their Information and Communication Technology (ICT) usage and readiness for online skill development programmes. The national statistics align with our findings, according to which more than 95% of our sample have access to the internet and electronic gadgets. Due to social distancing practices and lockdowns, the COVID-19 epidemic increased the usage of digital technology among Malaysians and normalised internet-based education among educators and learners. With the advent of the fourth industrial revolution, Malaysia’s education has heralded a new era of the learning system, referred to as Education 4.0. Education 4.0 is the latest approach to learning that equips learners with the skills needed for lifelong learning. It emphasises innovative teaching and learning methods and incorporates ICT in its processes. Education 4.0 encourages self-directed and more flexible learning at any time and location, facilitated by easily accessible ICT platforms and tools [40]. Highlighting the potential for utilising the internet to deliver the ToyBox Study modules to the preschool teachers in Malaysia, the eToyBox platform may assist teachers in their everyday work and curriculum implementation by utilising resources and tools now readily available in kindergartens.

### 4.1. Development and Evaluation of eToyBox

The development was divided into three stages: (i) planning of production, (ii) production of the materials, and (iii) evaluation. The first stage began with planning to produce eToyBox educational materials. Commencing with the initial stage, the focus was directed towards planning the eToyBox educational materials. Workshops, overseen by the research team, were conducted to delineate the learning goals for each module and ensure congruence between the online materials and the intended objectives. Decisions regarding material selection within each module were contingent upon whether the objectives pertained to knowledge acquisition, skill development, or attitudinal transformation.

To facilitate the teachers in achieving the learning objectives, a diverse range of educational materials was produced in the second stage, encompassing infographics, interactive games, storybooks, animations, and instructional videos. The production of these materials was characterised by careful attention to addressing various dimensions of the learning process. For instance, within unit 1, the strategic employment of infographics and interactive games sought to promote a better understanding, nurturing positive attitudes towards energy balance-related behaviour. Conversely, in units 2 and 3, the inclusion of demonstration videos, infographics, and storybooks aimed to support skill acquisition and foster familiarity with the integration of recommended classroom activities. It is worth noting that each education material within a unit could address multiple learning objectives, and various types of materials could be aligned with the same objective, which ultimately aimed to inspire teachers to implement the programme with enthusiasm and passion. Additionally, a short assessment was devised to gauge teachers’ comprehension of the materials upon completion of each module.

Upon mapping all module learning outcomes to their corresponding materials, the production phase commenced as the second stage, focusing on two crucial aspects: visual design and content delivery. Concerning visual design, all materials comprising elements such as video footage, animations, illustrated images for slide-based presentations, infographics, storybooks, and interactive games underwent meticulous review by the research team. This aimed to ensure accuracy, prevent potential misinformation or confusion, and ascertain their suitability for the local community. In addition, the team engaged in discussions and utilised preliminary pictograms to outline the content for infographics. These pictograms served to plan and organise the material’s interface on the online learning platform.

To ensure consistency and accuracy in content delivery, collaboration between the research team and a graphic illustrator was fostered to discuss the storyboard and script of animations and demonstration videos. Additionally, materials like infographics, interactive games, and stories were thoroughly reviewed by the research team to enhance the readability and comprehensibility of the modules. During the content development process, careful consideration was given to the literacy levels of preschool teachers, with a conscious effort made to avoid jargon-heavy language, thereby promoting accessibility. Following the finalisation of the learning materials, the subsequent step involved designing the online learning platform, which serves as the designated learning management system for delivering instructional content. This phase involved productive discussions among team members and web developers, supported by rough sketches or outlines that outlined the site’s architecture and visual organisation.

The phase three evaluation process was further subdivided into content validation and functionality evaluation. To ensure that the content of the online materials was actionable, readable, and understandable, feedback was collected from external experts in healthcare, sports education, and child education. Their expertise and insights played a crucial role in identifying areas for improvement. Additionally, the perspectives of kindergarten teachers were gathered to assess the acceptability and effectiveness of the demonstration videos in their teaching practices. Incorporating the feedback received, revisions were undertaken to enhance the materials. Content reorganisation was employed to ensure better understanding by the targeted audience. For instance, the storybooks were revised to simplify the storylines and dialogues, aligning them with the comprehension level of preschoolers. The iterative process of gathering feedback and making necessary revisions ensured that the materials underwent refinement and optimisation to improve their maximum effectiveness in meeting the needs and expectations of both teachers and young learners.

Additionally, the research team for ToyBox Study Malaysia conducted an evaluation of the learning platform’s functionality to ensure that the features available in eToyBox were functioning as intended. To guarantee compatibility during pilot testing, the team performed tests on various devices such as smartphones, tablets, and laptops, as well as across different browsers like Google Chrome, IOS Safari, and Microsoft Edge. The team assigned various learning activities to be completed on the online learning platform across the four modules. Subsequently, an open-ended survey was administered to gather feedback. The survey included questions addressing technical and functional aspects, such as identifying any bugs that affected feature performance or highlighting any concerns about the placement of graphics or text that influenced readability. Based on the feedback received, the design of the online learning platform for eToyBox was refined in preparation for the pilot test involving the Malaysia Community Development Department (KEMAS) preschool teachers.

### 4.2. Expected Outcome and Impacts of eToyBox

Overweight children and children with obesity have a higher probability of remaining obese into adulthood and having higher risk of developing non-communicable diseases [41]. Therefore, the readiness of preschool teachers to teach healthy habits in kindergartens should be prioritised as an initial strategy to instil awareness and proper healthy lifestyle behaviour among children because addressing childhood weight management issues at an early age is evidenced to reduce morbidity and mortality associated with obesity and long-term healthcare cost [42]. This study helps raise the public and government’s awareness of the significance of preparing educators to teach health-related subjects in kindergartens to address Malaysian children’s health issues. Furthermore, eToybox can bring support for healthy energy balance behaviours directly to the teachers, into kindergartens and homes, to encourage families to be active and eat healthily, and prevent or reduce obesity.

## 5. Discussion

The ToyBox Study Malaysia programme provides opportunities for preschool children to learn healthy eating and drinking habits and lead an active lifestyle. By instilling appropriate energy balance-related behaviours at an early age, these modules pave the way for a lifetime of healthy choices and wellbeing. The modules have been developed to be disseminated and taught by trained preschool teachers and incorporated into their usual classroom teaching sessions. While past studies evaluating the effectiveness of school-based nutrition education programmes have been limited to school-age children in Malaysia, the effectiveness of nutrition education intervention programmes is evident through the improvement of nutrition status and energy-related behaviour among participants [43]. Hence, preliminary findings suggest that ToyBox Study Malaysia has successfully filled this research need, as there were improvements in the four key behaviours (drinking, snacking, physical activity, and sedentary behaviour) as well as adiposity status (BMI z-score).

Despite this, it is evident that nutrition-related topics are still not systematically delivered in schools, as teachers face barriers such as a lack of availability of educational tools and insufficient knowledge on this topic [44]. Therefore, the ToyBox Study Malaysia programme fills this gap, particularly for preschool teachers, as it offers suitable education materials and lesson plans that complement the school syllabus, helping teachers teach nutrition topics more effectively in the classroom. Converting physical modules into an online learning management system, such as eToyBox, offers several benefits, including the ability to deliver professional development opportunities to early educators asynchronously. This approach is highly advantageous as it can be designed to reach a large geographic area and audience at their convenience.

Additionally, online interventions can be customised to incorporate interactive multimedia tools and other educational resources, which can greatly enhance participants’ engagement and interaction through various feedback functions available in the online learning system, ultimately enhancing the overall effectiveness of the intervention [45]. One of the most important aspects of the evolution of web-based intervention is the ability to obtain and analyse more accurate and non-biased data in real time with regular updates that can be susceptible to limitations such as inaccuracies in memory recall and social desirability bias [46]. It provides crucial insights into the programme’s effectiveness, allowing for timely adjustments to maximise its impact and achieve the desired outcomes. 

Like many other parts of the world, the COVID-19 pandemic has resulted in significant disruptions across various sectors in Malaysia, including education. The outbreak highlighted the importance of sustaining education programme delivery to ensure the continued dissemination of knowledge and maintain consistency in content delivery across diverse geographic locations whilst being sustainable and resilient and adapting to unforeseen disruptions and challenges. In response to the sudden interruption of academic activities, remote learning or web-based learning emerged as a sustainable practice to mitigate the impact of the disruption [47]. As the education sector has evolved and adapted to incorporate online learning in the wake of the pandemic, eToyBox has found its place within the relatively new hybrid learning approach. As well as offering resistance to future pandemic disruption, this approach has the potential to offer a more flexible and engaging learning experience, when compared to traditional fully online or fully on-site instruction [48].

By offering teachers access to additional nutrition education content beyond their teaching syllabus, it provides them with the support and guidance needed to improve their self-efficacy in teaching nutrition in schools [49]. As a result, it increases the likelihood of teachers being willing and motivated to teach this subject in the classroom [50]. Therefore, an online learning management system like eToyBox can provide professional development programmes and teaching materials on energy balance-related topics to empower teachers as agents of change to disseminate fundamental nutrition education to all.

## 6. Conclusions

ToyBox Study Malaysia has successfully transformed its nutrition education programme for preschool teachers from physical training to online learning, providing an alternative platform for delivering and learning about the content and delivery methods. This process involved adapting the European-based education modules to suit the Malaysian context. By converting printed modules into an online learning platform, preschool teachers in Malaysia now have increased opportunities to access evidence-based materials remotely. Moreover, this novel educational approach has the potential to improve the cost-effectiveness of childhood obesity-related intervention programmes, as it can be amplified to a larger audience through the use of the online platforms, enhancing its potential to promote healthy lifestyles among preschoolers and their families, further emphasizing the importance of a holistic approach to combating childhood obesity in Malaysia.

## Figures and Tables

**Figure 1 ijerph-20-06614-f001:**
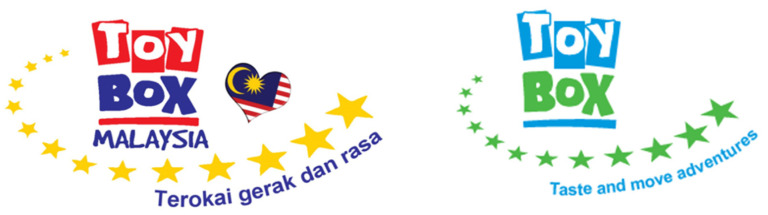
Adapted ToyBox Study Malaysia logo and the logo of the original ToyBox-study.

**Figure 2 ijerph-20-06614-f002:**
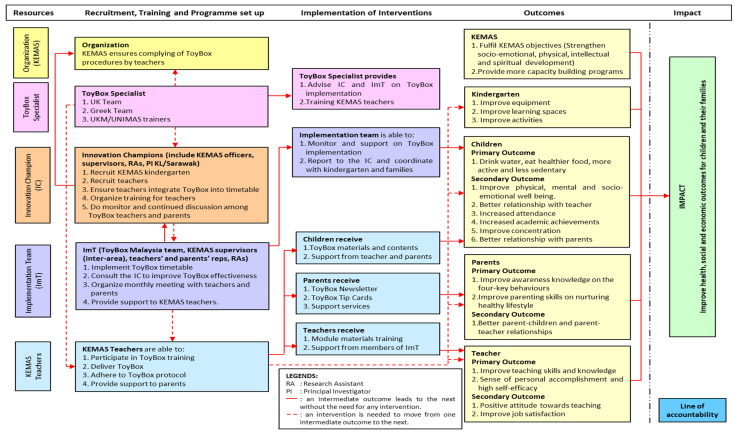
Theory of Change (TOC) map for the implementation of ToyBox Study Malaysia.

**Figure 3 ijerph-20-06614-f003:**
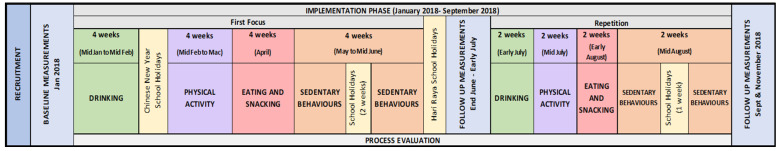
Implementation timeline for the ToyBox Study Malaysia intervention, including baseline and outcome evaluation.

**Table 1 ijerph-20-06614-t001:** Comparison of energy balance-related habits between baseline and post-intervention in Intervention and Control groups (mean ± SD).

Energy Balance-Related Habits	Intervention Group			Control Group		
	Baseline	Post-Intervention	Changes	Baseline	Post-Intervention	Changes
Drinking plain water (glasses/day)	3.86 ± 1.75	3.90 ± 1.90	+0.03	3.96 ± 1.84	3.80 ± 1.94	−0.16
Consumption of sweet snacks (portions/day)	0.82 ± 1.30	0.75 ± 0.97	−0.07	0.76 ± 1.22	0.79 ± 1.15	+0.03
Outdoor active play (minutes/day)	56.83 ± 71.64	61.33 ± 67.10 ^††^	+8.61	45.19 ± 61.47	43.94 ± 50.73	−1.24
Sedentary screen time (Screen time) (hours/day)	2.67 ± 2.17	2.72 ± 1.93	+0.05	2.46 ± 1.79	2.85 ± 2.03 *	+0.39

* Significantly different between baseline and post-intervention at *p* < 0.05 using paired *t*-test. ^††^ Significantly different between intervention and control group at *p* < 0.01 using independent *t*-test.

## Data Availability

The data presented in this study are available from the corresponding author upon reasonable request. The data are not publicly available due to privacy concerns.

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
