# Peer review of "From ToyBox Study to eToyBox: Advancing Childhood Obesity Reduction in Malaysian Kindergartens"

_ijerph, 2023, doi:10.3390/ijerph20166614_

Round 1

Reviewer 1 Report

This is a very interesting adaptation of the Toy Box study that will be of interest to early years and public health interventionists and researchers but across the paper there is a lack of clarity about what is being presented. For instance, the title is about the e-ToyBox but the materials presented are largely about the in-person 2018 portion of the study. I began to wonder if this was all published (although it was not obvious) and all of the sections about development and implementation and impact and outcomes and process evaluation were background to a study about the e-Toybox. If this is the case the methods, data collection and analysis used to learn about the e-Toybox need to be there and again, are minimal. I think the paper has to be tightened up from the title, through stating a purpose and objectives, to presenting section headings that add clarity - like a Methods section with measurement sub-sections for the outcome measures and the process evaluation measures, then a Results section (currently there are results presented in the paragraph about the measurement tool - which is minimally described- and then some process evaluation results (stakeholder satisfaction) then a description of a process evaluation that is being published elsewhere).  For example 3.0 Methods, 3.1 Outcome measures, 3.2 Process evaluation, 4.0 results.  Additionally, there are no defining headings delineating between presenting the 2018 face-to-face results and the e-Toybox study and methods. The information provided about the e-Toybox study doesn't seem like a protocol paper but it is very limited re: when is this adaptation being tested, what did you do to test it, what is the timeline. What methods were used to inform what you know about that adaptation?

So... as I said this is a very worthwhile endeavour - With the content that is included I would almost suggest changing the title and that the primary objective is to describe the development, implementation and implementation and health outcomes of  the Malaysian Toybox adaptation pilot (from development through to impact) and then  a secondary objective of the paper is to describe the adaptation to an e-Toybox. I leave the authors to decide what the intent of the paper is and to clarify this.

Author Response

We were delighted to be asked to contribute to this special issue. Thank you for your kind comments and suggestions, please see how we have responded to each comment below:

Response to reviewer 1

No

Reviewer’s Comments

Response

1

This is a very interesting adaptation of the Toy Box study that will be of interest to early years and public health interventionists and researchers but across the paper there is a lack of clarity about what is being presented.

For instance, the title is about the e-ToyBox but the materials presented are largely about the in-person 2018 portion of the study. I began to wonder if this was all published (although it was not obvious) and all of the sections about development and implementation and impact and outcomes and process evaluation were background to a study about the e-Toybox. If this is the case the methods, data collection and analysis used to learn about the e-Toybox need to be there and again, are minimal.

Thank you for your comments.

On your advice we have changed the title to: “From ToyBox Study to eToyBox: Advancing Childhood Obesity Reduction in Malaysian Kindergartens”.

We have previously published some specific aspects of the ToyBox Study Malaysia see for example Reeves et al., 2018; Lee et al., 2023 and Cheah et al., 2023, however this is the first time we have presented results from ToyBox Study Malaysia and introduced eToybox.

Besides, the development of eToyBox is now explained in greater details (please refer to section 4.1 Development and evaluation of eToyBox, page 11 to 12).

2

I think the paper has to be tightened up from the title, through stating a purpose and objectives, to presenting section headings that add clarity - like a Methods section with measurement sub-sections for the outcome measures and the process evaluation measures, then a Results section (currently there are results presented in the paragraph about the measurement tool - which is minimally described- and then some process evaluation results (stakeholder satisfaction) then a description of a process evaluation that is being published elsewhere).  For example 3.0 Methods, 3.1 Outcome measures, 3.2 Process evaluation, 4.0 results.

We appreciate the suggestion given. We have made amendments to the manuscript, please refer to the newly added sub-sections as follows:

2.0 Methods (line 95, page 2)

2.1 Approval for the ToyBox Study Malaysia (line 96, page 2)

2.2 Development of modules (line 221, page 3)

2.3 Theory of change intervention mapping (line 259, page 3)

2.4 Implementation (line 1, page 6)

3.0 Intervention impact and outcome measures (line 1, page 8)

3.1 Process evaluation (line 223, page 10)

3.2 Sustainability of obesity prevention intervention (line 232, page 10)

4.1 Development and evaluation of eToyBox (line 318, page 11)

3

Additionally, there are no defining headings delineating between presenting the 2018 face-to-face results and the e-Toybox study and methods.

The previous suggestion was taken. Corrections has been made to add sub sections accordingly as stated above.

4

The information provided about the e-Toybox study doesn't seem like a protocol paper but it is very limited re: when is this adaptation being tested, what did you do to test it, what is the timeline. What methods were used to inform what you know about that adaptation?

The development of eToyBox is now explained in greater details (please refer to section 4.1 Development and evaluation of eToyBox, page 11 to 12).

5

So... as I said this is a very worthwhile endeavour - With the content that is included I would almost suggest changing the title and that the primary objective is to describe the development, implementation and implementation and health outcomes of  the Malaysian Toybox adaptation pilot (from development through to impact) and then  a secondary objective of the paper is to describe the adaptation to an e-Toybox.

We have decided to change the title of our manuscript to “From ToyBox Study to eToyBox: Advancing Childhood Obesity Reduction in Malaysian Kindergartens” and updated the objectives of the paper as suggested. The primary objective describes the development, implementation, and intervention outcomes of the Malaysia ToyBox physical program. Additionally, the second objective is dedicated to detailing the adaptation process to eToyBox. The objectives are added at line 90, page 2.

Reviewer 2 Report

Reeves et al., aimed to describe an intervention to promote healthy lifestyles in preschool children by adapting the ToxBox tool in Malaysia. Additionally, the authors described the development of the tool in an online version. Interventions aiming to promote healthy lifestyles in preschool are critical to face childhood obesity and to improve health in later stages of life.  Regarding the present manuscript I have the following comments:  

Major comments

1.    In Figure 1 the authors mentioned that the impact would be “Improve health, social and economic outcomes…” Could you mention how this intervention improves economic outcomes?

2.    The authors mentioned a control group. Could you give more details about it? Did they have a different intervention or no intervention at all? How did you ensure that the ECE in the control and intervention groups were similar? 

3.    The authors mentioned that the intervention helped children to become more active during school days. Considering the questionnaire was applied to parents. Was this increase in physical activity during the afterschool time? Please explain.

4.    Were there specific aspects of the original project (European Version) that were more difficult to implement in Malaysia? Please mention which ones and what did you do to overcome this.

5.    It would be interesting if the authors could report the prevalence of meeting the recommendations for physical activity, sedentary time before and after the intervention.   

6.    I am not sure if is necessary to include the information about the eToyBox. All the results the authors show are after the traditional intervention before Pandemic. The authors don’t report any results after the implementation of the eToyBox. Please reconsider the title of the manuscript and rethink what the eToxBox adds to the present work. As I see it seems like two different manuscripts. The first one reports the results of the implementation of the ToxBox. The second one describes the development of a eTool with no results (yet). Both are very interesting but I don’t think that it’s a good idea to put them together in the same manuscript.

7.    Most part of the discussion focused on the utility of the eToxBox. Considering the comment above, the discussion should be focused on the implementation and results of the traditional implementation of the ToxBox.   

Minor comments

1.    I recommend using 5 years old instead of using 5's.

2.    I recommend using softer colors for the figures and more contrast with the letters (Figure 2 and 3). Also, be sure to improve the quality of the image (Figure 2).

3.    Figure 2:

a.    Please simplify the figure.

b.    Please include the abbreviations meaning in the legend of Figure 2.

c.     Reduce the number of words if possible.

d.    Please include the meaning of the different arrows if they have any. Straight and dotted arrows.

Author Response

We were delighted to be asked to contribute to this special issue. Thank you for your kind comments and suggestions, please see how we have responded to each comment below:

Response to Reviewer 2

Number

Reviewer’s Comments

Response

1

In Figure 1 the authors mentioned that the impact would be “Improve health, social and economic outcomes…” Could you mention how this intervention improves economic outcomes?

Economy is listed as one of the impacts in the figure, and this is clarified in the text were we report how obesity has substantial economic healthcare cost implications for nations and long term health care costs, reference (Ling, J. et al 2022)

2

The authors mentioned a control group. Could you give more details about it? Did they have a different intervention or no intervention at all? How did you ensure that the ECE in the control and intervention groups were similar? 

The control group did not receive any form of intervention during and after the study.

Please refer to the correction highlighted at line 22, page 6.

3

The authors mentioned that the intervention helped children to become more active during school days. Considering the questionnaire was applied to parents. Was this increase in physical activity during the afterschool time? Please explain.

The change in time spent on physical activity are reported under sub-section 3.0 Intervention Impact and Outcome Measures. (Please refer line 12-13, page 8 and Table 1 on page 11)

4

Were there specific aspects of the original project (European Version) that were more difficult to implement in Malaysia? Please mention which ones and what did you do to overcome this.

We appreciate the suggestion given. The challenges faced during the process of adaptation of the ToyBox Malaysia program has been added under the sub-section 2.4 Implementation. Please refer to line 25-41, page 6.

It describes the difficulties encountered when adapting the physical games and excursions from the original module to the local context due to space constraints issue in the preschool premises. Besides that, the adoption of the Malaysia Healthy plate from the Magic Snack Plate provides more information on healthy food choices and appropriate portion sizes.

5

It would be interesting if the authors could report the prevalence of meeting the recommendations for physical activity, sedentary time before and after the intervention. 

We have clarified this result. Please refer to section 3.0 Intervention impact and outcome measures. (Please refer line 12-13, page 8 and Table 1 on page 9)

6

I am not sure if is necessary to include the information about the eToyBox. All the results the authors show are after the traditional intervention before Pandemic. The authors don’t report any results after the implementation of the eToyBox.

Please reconsider the title of the manuscript and rethink what the eToxBox adds to the present work. As I see it seems like two different manuscripts. The first one reports the results of the implementation of the ToxBox. The second one describes the development of a eTool with no results (yet). Both are very interesting but I don’t think that it’s a good idea to put them together in the same manuscript.

We have taken in the suggestion from Reviewer 1 to change the title of the manuscript to “From ToyBox Study to eToyBox: Advancing Childhood Obesity Reduction in Malaysian Kindergartens” . By modifying the manuscript title, the primary focus is now on describing the development, implementation, and intervention outcomes of the Malaysia ToyBox physical program. Additionally, the second objective is dedicated to detailing the adaptation process to eToyBox. The objectives are added at line 90, page 2.

7

Most part of the discussion focused on the utility of the eToxBox. Considering the comment above, the discussion should be focused on the implementation and results of the traditional implementation of the ToxBox.

The results for ToyBox Malaysia program is updated in subsection 3.0 Intervention impact and outcome measures. Please refer to line 4 to 13, page 8.

8

I recommend using 5 years old instead of using 5's.

Please refer to line 41, page 1

9

I recommend using softer colors for the figures and more contrast with the letters (Figure 2 and 3). Also, be sure to improve the quality of the image (Figure 2).

Figure 2 and Figure 3 have been updated to feature softer colors for improved visual appeal. Please refer to figure 2, page 5 and figure 3, page 7.

10

For Figure 2:

a.    Please simplify the figure.

b.    Please include the abbreviations meaning in the legend of Figure 2.

c.     Reduce the number of words if possible.

d.    Please include the meaning of the different arrows if they have any. Straight and dotted arrows.

The text in figure 2 has been simplified; legend is added at the bottom of the figure to depicts the meanings of the arrows.

Please refer to figure 2, page 5.

Round 2

Reviewer 2 Report

Most of my comments were answered by the authors. I just have some minor additional comments. 

1. It should be discussed any possible reasons why physical activity increased more in the control group than in the intervention (10 min more in the control group vs just 4 min in the intervention group) Could the authors discuss some explanations for this? 

2. In Table 1. Are the data expressed in median (Interquartile Range) or in mean (SD)?  Please verify

Author Response

Thank you for your kind comments on our most recent submission, please see our responses in the table below.

No

Reviewer’s Comments

Response

1

It should be discussed any possible reasons why physical activity increased more in the control group than in the intervention (10 min more in the control group vs just 4 min in the intervention group). Could the authors discuss some explanations for this?

We thank the Reviewer for this insightful question. We have decided to report the mean (SD) values instead of the median (IQR) values. As such, we have now clarified the results and provide a clearer explanation. Please see page 9 paragraph 1.

2

In Table 1. Are the data expressed in median (interquatile range) or in mean (SD)? Please verify.

The original Table 1 presented the data as median (IQR). However, following your query we have now presented the mean (SD) results to add clarity to the revised Table 1. (See page 10)